# Effects of Genetic Diversity on Health Status and Parasitological Traits in a Wild Fish Population Inhabiting a Coastal Lagoon

**DOI:** 10.3390/ani15152195

**Published:** 2025-07-25

**Authors:** Alejandra Cruz, Esther Lantero, Carla Llinares, Laura Ortega-Díaz, Gema Castillo-García, Mar Torralva, Francisco J. Oliva-Paterna, David H. Fletcher, David Almeida

**Affiliations:** 1Facultad de Medicina, Universidad San Pablo-CEU, CEU Universities, Urbanización Montepríncipe, 28660 Boadilla del Monte, Spain; alejandra.cruzvarona@ceu.es (A.C.); esther.lanterobringas@ceu.es (E.L.); carla.llinaressimon@usp.ceu.es (C.L.); laura.ortegadiaz@usp.ceu.es (L.O.-D.); gcastillo@cnb.csic.es (G.C.-G.); 2Department of Plant Molecular Genetics, Centro Nacional de Biotecnología—CSIC, Campus Universidad Autónoma de Madrid, 28049 Madrid, Spain; 3Department of Zoology and Physical Anthropology, University of Murcia, 30100 Murcia, Spain; torralva@um.es (M.T.); fjoliva@um.es (F.J.O.-P.); 4UK Centre for Ecology and Hydrology, Environment Centre Wales, Bangor LL57 2UW, UK

**Keywords:** black-striped pipefish, body condition, disruptive selection, Iberian Peninsula, life-cycle complexity, Mediterranean Sea

## Abstract

Parasites impose selective pressures on wild fish, depending on host genetic variability. Coastal lagoons are transitional ecosystems where no study exists on fish genetic diversity and their parasites. Black-striped pipefish were collected in summer from the Mar Menor (a Mediterranean lagoon, SE Spain). The frequency of individuals with a medium level of genetic diversity was lower in the sampled population. For this same category, both internal and external health indices indicated a worse status, as well as a higher number of parasitised fish. A particular type of natural selection appears to be acting: disruptive selection, where parasites exert a greater disease pressure against the genetically intermediate individuals. This study demonstrated two clear genetic strategies displayed by hosts to better control parasitism: (1) low diversity, and (2) high diversity. Both categories may reflect a stronger immune response. Thus, parasites can change genetic diversity within animal populations, which will affect the evolution of host species.

## 1. Introduction

Parasites can exert a strong selective pressure on host populations, from invertebrates to mammals, with profound consequences for the entire ecosystem and evolutionary processes [1,2,3]. This is because many parasite species promote diseases that can deeply impact health, disturbing a wide variety of traits related to host life cycles, such as growth rate, reproduction, behaviour or feeding habits [1,4,5,6]. Ultimately, parasites displaying an elevated pathogenic nature may massively decrease fitness until high levels of mortality are observed within the host populations. As an example, a dramatic case was found for the pen shell *Pinna nobilis*, reaching up to 100% in some sites [7]. In general, parasitism is an ecological relationship widespread across the globe that usually does not lead to such extreme consequences, such as mass mortalities [8]. Thus, data on physical condition, fitness or health status in hosts constitute valuable information in order to truly assess the sub-lethal effects of this biotic interaction at the population and ecosystem levels [9,10,11].

Among vertebrate hosts, fish are pivotal communities for most aquatic ecosystems all around the world, with many stocks being of great interest for exploitation [12]. Given that wild fish are host to a wide diversity of parasite taxa, with >30 k helminth species [13], these communities can be considered good biological models to investigate particular ecological/evolutionary patterns from such intimate relationships [14,15,16]. As an example of a simpler interaction, monogeneans are monoxenous gill infesting ecto-parasites through the branchial chamber [16]. More complex is the case of digeneans, which are heteroxenous parasites that typically include a variety of invertebrate and vertebrate hosts in their life cycle: molluscs, crustaceans, fish, birds or even aquatic mammals. Also, they have free-swimming stages or larvae during their transmission between hosts [14,17]. Thus, variations in abundances of particular parasite communities may be a result of changes in structure and dynamics at the ecosystem level [15,18].

The genetic aspect of this biotic interaction is of particular relevance for research on ecology and evolution, as previous studies have demonstrated genetically based variations for key traits affecting this relationship, from both the parasite and the host perspectives (see comprehensive reviews on this topic in [19,20]). Focusing specifically on animal health, hosts have evolved/developed different traits, or strategies with their particular genetic basis, to maintain fitness when interacting with parasites. More specifically, physiological/genetic mechanisms are usually related to [21,22] (1) the defence against pathogens/parasites by the immune response to avoid establishment or reduce burden, and (2) rapid body repair via cell/tissue regeneration from the damage caused by infections. Moreover, the link between the host genotype and the level of parasite infection appears to be highly variable. Indeed, the effect of genetic traits can be contrasting at the host population level, in terms of parasites mediating natural selection [23]: balancing, stabilising, directional or disruptive. According to the inbreeding depression hypothesis, a significant negative relationship is expected to be detected between host heterozygosity level and parasite burden [24]. This may be indicative that parasites contribute to modulate genetic diversity in their host populations through a combination of balancing and directional positive selection for the most heterozygous (i.e., genotypically diverse) individuals. Zueva et al. [25] observed this relationship between the Atlantic salmon *Salmo salar* and the ecto-parasite monogenean *Gyrodactylus salaris*. However, parasite-mediated disruptive selection can also occur, as evidenced by Blanchet et al. [26,27] in a wild population of common dace *Leuciscus leuciscus* and its ecto-parasite copepod *Tracheliastes polycolpus*. In these dace studies, parasite burden was lower for fish individuals displaying both extremely heterozygous and homozygous levels. This implies that selective pressure is exerted against phenotypically intermediate individuals [23], with global genetic variance or diversity of this population being broadened accordingly. Therefore, it is possible that different and alternative parasite-mediated mechanisms shape genetic diversity at the host population level. Clearly, there is a need for more research that provides insights into the relationships between genetic traits of hosts and their parasite infra-communities, particularly for fish.

In terms of applications, basic knowledge on genetic traits of wild parasitised fish could be interesting for aquaculture. Given the collapse of many extractive fisheries at the global scale [28], aquaculture is regarded as a feasible solution to provide finfish products, i.e., a source of high-quality protein, around one third of the current total consumption [28,29]. The world-wide development of this industry has led to an increasing awareness of the necessity to understand the genetic causes and consequences of parasite infections [30]. This is because many parasite species can result in harmful diseases and hence provoke major mortality outbreaks in farmed fish [31,32]. In fact, heritable genetic components can be identified to develop disease-resistant genetic lines tailored to particular target fish species through artificial selection [30,33]. Such programs of selective breeding, supported by genetic/genomic information, also contribute positively to the overall productivity and economic performance of aquaculture activities [34,35].

Coastal lagoons are considered transitional water bodies, located between flowing fresh waters (i.e., rivers, streams) and marine (salt-water) environments. These distinctive habitats have received relatively less attention than pure freshwater or marine ecosystems [36,37]. Regarding fish parasites in transitional waters, Giari et al. [38] reviewed the relevance of such communities for shaping structure/function within these ecosystems. Almeida et al. [39] have also provided valuable information on fish parasites from a Mediterranean coastal lagoon, although this study was focused on conservation purposes for the Iberian eco-region, with parasites being used as ecological indicators of habitat quality. Nonetheless, genetic data are still very scarce for these aquatic environments, beyond the use for mere identification or assessing parasite diversity [40,41]. To the best of the authors’ knowledge, information about diversity of fish genotypes related to their parasites has been virtually undescribed in coastal lagoons. Consequently, the assessment of these genetic and parasitological traits within resident populations of fish hosts, along with their health status, have a great potential to more effectively reveal ecological and evolutionary implications under these particular transitional environmental conditions. However, this potential is severely under-developed.

In light of this dearth of information, the aim of the present study was to assess the relationships among genetic traits, health status of fish hosts and their parasite infra-communities in a wild population inhabiting an Iberian coastal lagoon (Mediterranean Sea). Specifically, a variety of parameters were compared across host categories exhibiting different levels of genetic diversity: population frequencies and external/internal health indices calculated from fish data; and prevalence, taxonomic composition, abundances and an index of life-cycle complexity calculated from parasite data.

## 2. Materials and Methods

### 2.1. Study Area

The Mar Menor (smaller sea, in English) is a hypersaline water body (45 g L^−1^) separated from the Mediterranean Sea (38 g L^−1^) by a sandbar of 21 km length and 500 m width. This area is located in the south-east of the Iberian Peninsula (Murcia Region, Spain) (Figure 1). Mean values for the surface and depth of this coastal lagoon are 135 km^2^ and 4 m, with a maximum of 6 m. The climate of the study area is a semi-arid Mediterranean type, with intense summer drought (<5 mm) and rainfall concentrated in autumn–winter (20–30 mm). The average annual temperature typically ranges between 15 and 20 °C. The lowest temperatures occur in January (3 °C in air and 10 °C in water), while the maximum values are observed in August (>40 °C in air and 30 °C in water). A sedimentary cover of red and black silts from the Quaternary is the main substratum. The Ministry of Environment Spain (in Spanish: Ministerio para la Transición Ecológica y el Reto Demográfico, MITECO hereafter; [42]) provided data on climatic and geological features. Mixed meadows of seagrass *Cymodocea nodosa* and *Ruppia cirrhosa* are common in littoral shallow waters. Oliva-Paterna et al. [43] have previously described the diverse fish fauna of the Mar Menor, which acts as a nursery area. In terms of conservation, the Iberian toothcarp *Aphanius iberus* and the European eel *Anguilla anguilla* are examples of threatened species that are frequent within this coastal lagoon [44]. The Mar Menor is of great ecological importance, being an area with up to 10 approved categories of environmental protection. Some examples are (1) the Natura 2000 Network as Special Area of Conservation for Birds; (2) the Ramsar Convention as Wetland of International Importance; and (3) the Barcelona Convention as Specially Protected Area of Mediterranean Importance [45,46].

### 2.2. Field Sampling

Given that the particular target species (see below) spawns until July in the Mar Menor [39], fish were sampled after this breeding period in mid-August 2023 and 2024. Therefore, this sampling season avoids any effect of this physiological (i.e., reproductive) status on data. As mentioned, this was a two-year study, which allowed for the acquisition of a more representative dataset than that of a single-year study, according to Lanzoni et al. [47]. These authors recommend more than one year of sampling in transitional habitats, such as lagoons, to encompass all the variability present in fish populations from such dynamic ecosystems. According to hydrological data for the Mar Menor [42], both sampling years were approximately average. Therefore, the examined biological traits were considered representative. An appropriate spatial representativeness was accomplished by selecting six sampling sites all around the lagoon, which were evenly distributed and separated >5 km (shoreline) to minimise data dependence (Figure 1). Given the importance of the Mar Menor in terms of conservation (see above in Study Area), only those six sampling sites were allowed. However, this number of sites appeared to be sufficient to include the existing environmental variability, according to previous fish surveys by Almeida et al. [39].

Given that the aim of this study was to analyse a variety of biological traits in a wild fish population, specimens were sampled from sites under the nearest natural conditions. Thus, sampling sites were located in the vicinity of similar and undisturbed environmental surroundings, with regard to vegetation and shoreline structure. Consequently, sites under clear anthropogenic influences were avoided, such as agricultural inputs, towns and fishing areas. Tributaries were also avoided due to their potential effects on physico-chemical parameters, such as salinity or nutrient concentrations. Temperature, pH, salinity, dissolved oxygen, phosphates and nitrates were checked to ensure similarity across sites. Each site was randomly sampled on different dates for two weeks. Fish collection was always carried out at 10:00 (solar time) per site and weather conditions (e.g., air temperature, humidity, visibility) were similar among sampling dates. A consistent sampling protocol was followed by wading according to the European legislation (Comité Européen de Normalisation/International Organization for Standardization: CEN/ISO; [48]). Also, sampling effort was similar per site, in terms of time (1.5 h) and research team (the same three people, see Author Contributions below), to ensure data comparability. Similarly to Almeida et al. [39], the catch method consisted of hauling a beach-seine net (10 × 2 m, 2 mm mesh size) along a 20 m section of shoreline. Five replicates (i.e., 5 × 20 m section = 100 m shoreline) were collected at each sampling site. The area covered by each haul was approximately 175 m^2^ (total hauled area ≈ 900 m^2^ per site). At the meso-habitat scale, sandy-silt bottoms and patches of submerged vegetation were sampled to collect the target species: the black-striped pipefish *Syngnathus abaster* (pipefish, hereafter). Target species selection was a key point of the present research design for several reasons: (1) This small-bodied fish is easy to be transported and maintained in aquaria. (2) It is not under any threatened category, in terms of IUCN Red List, at the regional level in Murcia, and it is listed as Least Concern in the Mediterranean area [49]. (3) It is among the most abundant and common fishes in the Mar Menor coastal lagoon [43,44]. Consequently, the extraction of a relatively large fish sample (see *n* below in the Section 2.3.) had virtually no effect on this population, nor on subsequent data acquisition of the present two-year study. (4) Pipefish belongs to the taxonomic Order Syngnathiformes, being a zooplanktivorous species, swimming through aquatic vegetation at a certain distance from the bottom (i.e., limnetic habits) and inhabiting a coastal lagoon [50]. The phylogenetic position and environmental requirements of the study species are very contrasting with respect to previous studies analysing the connections between genetic diversity of hosts and fish parasites. Some examples are the case of a river leuciscid species in Blanchet et al. [26,27] or the Atlantic salmon in Zueva et al. [25]. Thus, the present study would contribute towards providing a more complete picture of this biotic relationship affecting host population traits, with parasites mediating directional, balancing or disruptive selection (see Section 4).

After fish collection at each site, specimens were identified. Tanks with cooling accumulators and portable aerators were used to hold pipefish and for transportation. Within the complete fish sample per site, the largest individuals were visually selected for laboratory examinations. To ensure collection of the largest size sub-sample, fish were measured for Standard Length (SL, ±1 mm). Such a selection was to minimise the effect of fish size on data. In fact, no significant differences were found between sites and genetic categories (see Section 2.6 and Section 3). Also, the use of this sub-sample ensures sufficient maturity to more effectively reveal health and parasitological patterns for small-bodied fishes (see a similar procedure in the work of Almeida et al. [39,51]). In particular for the Mar Menor, pipefish reach an age of up to 4 years [39]. Thus, the capture of larger/older specimens means that those individuals completed 2–3 reproductive cycles, minimising the effect of extraction on this wild population, according to the license for scientific fish sampling (see details below). In accordance with Näslund and Johnsson [52], all specimens were processed each day after pipefish collection at a laboratory located <1 h from the Mar Menor. This relatively short transportation period reduces animal stress and parasite disturbances/losses. Also related to fish captivity, these same authors recommend collecting samples of water, substrata and submerged vegetation from each site for the environmental enrichment of the aquarium at the laboratory (see details below).

All field procedures complied with animal use and care regulations of Europe and Spain (a specific license was granted for scientific field research in Murcia: AUF/2022–2024/0007–0083). Fish were collected by trained personnel, as the holder of the license was F.J. Oliva-Paterna. Thus, no adverse effects were caused on the wildlife in the study area and all native fish fully recovered from the collection methods. Except for the study species, fish were released after recovery at the same sites of the lagoon where they were captured.

### 2.3. Morphological Examination

A 50 L aquarium was equipped with a water filter, quality controllers for temperature and pH, and a pump to maintain a constant dissolved oxygen level, as well as an enriched environment according to each site [52]. This was an enclosed system, with no flow through. For an appropriate acclimatisation of pipefish, water was swapped between the tank and the aquarium for a 30 min period. Then, individuals were changed from the tank to the aquarium one by one. Following an adapted procedure from the work of Chapman et al. [53], clove oil was used to euthanise each fish specimen by immersion in a 0.5 L beaker with water from the aquarium (100 mL). This anaesthetic was added (50 µL every 20 s) with a micro-pipette, gently aspirating and releasing with the tip for a better mixture and dispersion of oil. This procedure progressively reduced the operculum movement until reaching a lethal overdose solution after 5 min, when a complete motionless body was observed. Then, the spinal cord was severed just behind fish head with fine scissors. SL was measured per individual. Among different measurements of fish length, SL avoids using the caudal fin, which can introduce errors from abnormalities on the rays or skin [54]. In total, *n* = 360 individuals (30 ind. × 6 sites × 2 years) were examined. The size range was 80–125 mm SL. After measuring body size, dissections were conducted to examine specimens, including sex determination.

Body Condition (BC) consists of calculating a mass index that accounts for the animal size and can be compared to other values from reference populations [55,56]. Therefore, BC can be considered an external morphological approach to measure fish health [39,57]. In this study, eviscerated Body Mass (eBM) was measured (electronic balance, ±1 mg) to provide an integrated quantification of BC, avoiding bias from gonad mass and gut content. The eBM range was 192–799 mg. Then, data on fish size (SL) were used for the calculation of BC (see details in the Section 2.6).

Adapted from the work of Adams et al. [9], a Health Assessment Index (HAI, hereafter) was computed as another morphological approach to estimate fish fitness. Data from several internal organs is necessary to calculate this index. Consequently, the health status of hosts is better assessed by using both BC and HAI, in terms of physiological responses related to environmental features. Also, parasite results are complemented by HAI data, with these (health and parasitological) traits being related to the resistance/tolerance capacity of fish hosts (see the Section 4). Specifically, gills and skin (fins, scales) were externally examined. Then, a variety of internal organs were examined during dissections: lens of the eye, cardiac muscle, intestines, hepatic lobes, splenic pulp and testis/ovaries. Abnormalities, pigmentations and hyperplasia/hypertrophy were specifically assessed. Scores were assigned to each of the eight anatomical categories, such that a maximum possible score of 30 points relates to poor status, whereas a minimum score of zero points relates to good status. Then, the HAI was calculated for each specimen by summing the eight scores (total range: 0–240 points), according to the following formula:$$HAI=\sum_{i}^{n}HS_{i}$$
where HS*_i_* is the Health Score of the anatomical region/organ *i* and *n* = 8 is the total number of examined anatomical regions/organs on a particular fish individual.

### 2.4. Parasitological Examination

Parasites were observed and extracted during HAI dissections. Identification and counting were carried out for external and internal specimens from each host individual. Previous studies have examined a relatively low number of fish hosts, although these were representative enough after applying curves of species accumulation. As an example of this, Figure 1 in the work of Chapman et al. [53] shows <20 fish individuals per site. Consequently, the sample size in the present work (*n* = 30 fish individuals per site) achieved a good representativeness of parasite infra-communities for pipefish (see other examples of this procedure in the works of Cruz et al. [57] and Almeida et al. [39,51]). Standard protocols [58,59] were applied during fish examinations. Parasites were detected by using dissecting (40×) and phase-contrast (1000×) microscopes. Specialised keys [60,61,62,63] were used for species identification, except for some taxa where Family or Genus categories were the maximum levels reached. In these cases, existing information was enough to estimate their pathological effects and assign the number of different host species within their life cycles [53,64]. Excystation with fine forceps, fixation in ethanol and staining were occasionally necessary to identify parasites. Scientific names used for parasites are currently accepted, according to the Global Biodiversity Information Facility (GBIF; [65]).

### 2.5. Genetics

Total DNA was extracted and quantified from a muscle sample (≈25 mg, right body flank) [66]. DNeasy^®^ Blood & Tissue Kit (Qiagen, Valencia, CA, USA) and Nanodrop^®^ Lite Spectrophotometer (ThermoScientific, Wilmington, DE, USA) were, respectively, used for these two procedures. Based on previous genetic analyses in *Syngnathus* species [67,68], highly polymorphic microsatellite *loci* were used (*n* = 10, Table 1), also showing amplification under similar conditions. Microsatellite markers are often used as a surrogate for genome-wide diversity to infer associations between parasite load and host genetic traits [24], including in fish [26,27,69]. Polymerase Chain Reactions (PCRs) co-amplified these markers from 20–25 ng of gDNA (16 μL) in 8 μL of 2 × QIAGEN Multiplex PCR Master Mix (Qiagen, Valencia, CA, USA), along with a *locus*-specific optimised combination of primers (detailed protocols are available from the authors upon reasonable request). PCR were performed in a Sure Cycler 8800 machine (Agilent Technologies, Santa Clara, CA, USA): 15 min (95 °C) + 30 cycles × [30 s (94 °C) + 90 s (56 °C) + 60 s (72 °C)] + 45 min (72 °C). The last time period was for a final step of elongation. After amplifications, ABI PRISM 3730 (Applied Biosystems, Foster City, CA, USA) was used to separate fragments by capillary electrophoresis in the Department of Genomics/Proteomics at the Complutense University (Madrid, Spain). Allelic sizes were scored using the MW marker GeneScan-500 LIZ^®^ (Applied Biosystems, Foster City, CA, USA). Genotypes were analysed by using the software Peak Scanner v.2.0 (Applied Biosystems, Foster City, CA, USA) to generate individual profiles with the 10 microsatellites.

### 2.6. Data Analyses

To improve the normality and homoscedasticity assumptions, log-transformation was used on data: log_10_ (*x* + 1). A Shapiro–Wilk test was applied to check that values were fitted to normal-shape frequencies. Levene’s test was used to verify homoscedasticity. The software R v.4.1.0 [70] was used to perform statistical analyses. The significance level was set at α = 0.05. The values reported in results are untransformed arithmetic Means ± Standard Deviations (SD).

With respect to genetic traits, estimates of null alleles were calculated using Arlequin software v3.5.1.3 [71], with the Expectation–Maximisation (EM) algorithm described by Dempster et al. [72]. Linkage Disequilibrium (LD) coefficients (D, D’ and *r*^2^) were calculated between pairs of alleles at different microsatellite *loci* also using Arlequin with 10,000 permutations. No null alleles or LD were detected in the selected set of microsatellites. Two genetic indices of homozygosity were calculated using the Package GENHET v.3.1 in R software [73]: Internal Relatedness (IR; [74]) and Homozygosity by *locus* (HL; [75]).IR = (2H − ∑ *f_i_*)/(2N − ∑ *f_i_*),
where H is the number of *loci* that are homozygous, N is the number of *loci* and *f_i_* is the frequency of the *i*th allele contained in the genotype.HL = ∑ *E_h_*/(∑ *E_h_* + ∑ *E_j_*),
where *E_h_* and *E_j_* are the expected heterozygosities of the *loci* that an individual bears in homozygosis (*h*) and in heterozygosis (*j*), respectively.

As per Blanchet et al. [26], both indices were highly correlated (*r* = 0.986, *p* < 0.001), with only IR (range: from −0.125 to 0.470) being finally reported in the present study (see a similar genetic approach in the work of Cruz et al. [57]). This genetic measure (i.e., IR) attempts to estimate the relatedness of an individual’s parents using the extent of allele sharing relative to random expectations [74]. Therefore, IR was used as a measure of Genetic Diversity in the present study (similar to Blanchet et al. [26]). Additionally, IR often outperforms other indices when predicting individual fitness [76]. Individuals with low IR values are more heterozygous within the population, whereas high IR levels correspond to more homozygous specimens. Cruz et al. [57] and Blanchet et al. [26] revealed simple-linear and curvi-linear (i.e., quadratic) regressions, respectively, between IR and parasite burden of leuciscid fishes. As per Cruz et al. [57], data were fitted to linear and polynomial functions by using Generalised Additive Models (GAMs). This technique was performed because, unlike more conventional regression methods, it does not require the assumption of a particular shape for the relationship (e.g., linearity) between variables. The highest supported GAMs were selected using the Akaike information criterion [77]. However, no continuous association was found to be statistically significant (overall *p*-values > 0.05) for the present dataset (neither total nor endo-parasite abundances). Alternatively, and according to the IR range (total amplitude ≈ 0.6, see above), three categories of genetic diversity were arbitrarily established (interval ≈ 0.2 per category): low (IR > 0.3), medium (0.1 < IR < 0.3) and high (IR < 0.1) levels. Then, all fish data were distributed into each corresponding category to test for genetically based differences in population, morphological and parasitological traits (see the Section 3). This analytical technique (i.e., splitting IR into categories) was used to more clearly detect and visualise variation patterns (see a similar procedure in the work of Blanchet et al. [27]), as well as to ease biological interpretations.

General Linear Models (GLMs) were used to test for differences between males and females on the examined traits, as per Almeida et al. [51]. Specifically one-way Analysis of Variance (ANOVA) was used to test for differences between sexes (independent variable) for every morphological/parasitological parameter (dependent/response variables). Also, two-way ANOVA was used by including a second factor as Genetic Category (see above). Analysis of Covariance (ANCOVA) was applied with SL/eBM as the covariates to control the effect of size/mass. Given that no difference was found (overall *p*-values > 0.05), this categorical factor (i.e., gender) was not included in subsequent models. This allowed for the simplification of data analyses and thus increased the statistical power of the remaining sources of variation, which would otherwise be seriously compromised [39,51]. Data were pooled because no spatial/temporal effect was found to be significant for the dependent variables (see above), after the use of previous Generalised Linear Mixed Models (GLMMs) using site and year as the random factors (a comprehensive review of this statistical technique is given in the work of Johnson et al. [78]).

In order to reveal the genetic pattern at the population level, the numbers of collected fish individuals were used to calculate percentages per IR category. Parasite prevalence was the proportion of examined hosts infected with all parasites or specifically endo-parasites. This parameter was also calculated as a percentage across the three categories. Then, chi-squared tests were used to detect significant differences for these two response variables (i.e., population frequencies and prevalences) between genetic categories, along with a post hoc Marascuilo tests for pair-wise comparisons. Chi-squared tests were calculated from data on counts, but these results were presented as percentages for a clearer visualisation and interpretation.

One-way ANOVA was used to ensure the existence of highly significant differences in the IR means between the three genetic categories (see the Section 3). Then, a post hoc Tukey–Kramer Honestly Significant Difference (HSD) test was used to perform pair-wise comparisons across genetic categories. ANOVA was also used to confirm that similar fish sizes (SL) were selected between genetic categories (see the Section 3). For comparison purposes, the potentially confounding effect of fish length was avoided by using ANCOVA. For the BC assessment, eBM was compared among genetic levels with SL as the covariate. This analytical procedure, i.e., the use of body length as the covariate, is statistically preferable to control for size effects because it avoids the bias of using the residuals from linear regressions between the study parameters and the size [56] or computing indices/ratios (e.g., Fulton’s condition factor) [56,79].

Parasite infra-communities were compared among genetic levels of diversity by calculating quantitative parameters per host [51,80]: total Abundance (tA, counts from all parasite taxa) and a Life-cycle Complexity Index (LCI). The latter descriptor was calculated according to the following formula:$$LCI=\sum_{i}^{n}\left(a_{i}\times h_{i}\right)/\sum_{i}^{n}a_{i}$$
where *a_i_* is the abundance of a parasite taxon *i*, *h_i_* is the number of different hosts in the life cycle of *i* and *n* is the total number of parasite taxa on a particular fish individual. The number of hosts was allocated to each parasite taxon according to the available literature (see identification keys above). This number ranged from 1 to 3 hosts for the parasite taxa found in the present study and, consequently, the descriptor LCI (i.e., number of hosts weighed by the abundance of each parasite taxon) also varied within this range. Additionally, endo-parasite Abundance (endoA) was calculated per fish individual. This was because the effect of habitat conditions can differentially affect the parasitism type, with ecto-parasites being more exposed to the external environment, whereas endo-parasites inhabit fish internal environment under more stable conditions. Also, endo-parasites usually impact fish health more seriously, better reflecting the host genetic traits and the associated immune response (see the Section 3 and Section 4). As with the procedure for BC, ANCOVA (covariate: eBM) was used to test for significant differences between genetic categories for the response variables HAI and the three parasite descriptors (tA, LCI and endoA).

## 3. Results

Fish sizes were similar across the three genetic categories (ANOVA: *F*_2,357_ = 0.66, *p* = 0.518), with SL results (mm) as follows: low = 93.9 ± 15.32, medium = 89.9 ± 12.85, and high = 91.4 ± 14.56 levels. After controlling for fish size (SL), adjusted eBM values (i.e., BC) were lower (mean < 315 mg) for the medium level in comparison with the low and high levels (>400 mg), although this difference did not reach statistical significance (but *p* < 0.1 from ANCOVA; see Table 2). After controlling for fish mass (eBM), adjusted HAI means were very different among the genetic groups (ANCOVA: *p* < 0.01; see Table 2). In particular, this health index was significantly higher (>35 points) for the medium level, whereas the low and high levels were similar (≈25 points) (Table 2).

The calculation of population frequencies showed a lower percentage of fish individuals displaying a medium level of genetic diversity (26.7%, *n* = 96), in comparison with the low (35.8%, *n* = 129) and the high (37.5%, *n* = 135) levels (Figure 2). However, this difference was statistically non-significant (*χ*^2^ = 5.65, *p* = 0.059).

Pipefish showed a higher percentage of parasite prevalence for the medium level of genetic diversity (74.0%, *n* = 71) relative to the low (65.9%, *n* = 85) and high (62.2%, *n* = 84) levels (Figure 3), although this difference was not statistically significant (*χ*^2^ = 3.53, *p* = 0.171). This pattern was similar for endo-parasites, with a higher prevalence for the medium level (67.7%, *n* = 65). On this occasion, such differences did reach the statistical significance level (*χ*^2^ = 6.60, *p* = 0.037). The difference in prevalence was close to the statistical significance between the medium (67.7%) and the low (55.0%, *n* = 71) levels (post hoc Marascuilo: *p* = 0.054), and clearly significant between the medium (67.7%) and the high (51.1%, *n* = 69) levels (post hoc Marascuilo: *p* = 0.012) (Figure 3).

A wide variety of parasites were detected on fish hosts, belonging to 13 different taxa (Table 3). Digeneans were the group showing the highest taxonomic richness (six categories) and abundance (>65% from the three genetic levels). In particular, the most abundant digenean parasites were the taxonomic Family Cyathocotylidae, followed by *Podocotyle atherinae*, *Cryptocotyle concava* and the Genus *Diplostomum*. For all these parasites (as well as the rest of digeneans, the nematode *Contracaecum microcephalum* and the cestode *Proteocephalus* sp.), the highest abundances were always found in the medium level of genetic diversity (Table 3). Other common parasites for pipefish were the copepod *Ergasilus ponticus*, the monogenean Genus *Gyrodactylus*, the acanthocephalan *Pomphorhynchus laevis* and the ciliate *Trichodina partidisci*, with these four parasite taxa being less abundant in the medium category. The digenean *Timoniella imbutiformis* was not observed in more heterozygous pipefish, whereas *Echinochasmus perfoliatus* (digenean) and *Paracanthocephaloides incrassatus* (acanthocephalan) were not detected in more homozygous hosts (Table 3).

Regarding the three parameters of parasite infra-communities (tA, LCI and endoA), the adjusted values (covariate: eBM) were always higher (>20 parasite specimens and ≈2 host species, respectively, per parameter) for the medium group relative to the low and high levels (≈18, ≈1.6 and ≈12, respectively, per parameter) of genetic diversity. Such differences were close to statistical significance (ANCOVA: *p* < 0.08; see Table 4) for tA and were clearly significant (ANCOVA: *p* ≈ 0.02; see Table 4) for LCI and endoA. In particular for these two parasite indices, the medium group was significantly different with respect to the low and high levels (Tukey’s HSD test: *p* < 0.05; see Table 4).

Genetic diversity (measured as IR, in an inverted order) significantly varied across the three categories (ANOVA: *F*_2,357_ = 106.22, *p* = 0.000007), with all the groups being different (Tukey’s HSD test: *p* = 0.0008). The specific IR results were as follows: low = 0.369 ± 0.0712, medium = 0.189 ± 0.0758 and high = 0.005 ± 0.0684 levels.

## 4. Discussion

Three categories of genetic diversity were established, according to IR values: low, medium and high levels. Several biological traits varied depending on these levels of homozygosity within the studied pipefish population. Regarding population frequencies, fewer individuals were of the medium level, although the difference was not statistically significant. In terms of morphology, BC and HAI were greater for the extreme levels of genetic diversity. These findings suggest a better health status for the most homozygous and heterozygous individuals, probably due to a higher tolerance capacity. However, this was statistically significant for HAI only. Parasite prevalence and abundance were higher for the medium IR level, although not significantly so. Such patterns were much clearer, statistically speaking, when only endo-parasites were accounted for. This suggests a poor resistance and worse immuno-competence for intermediate values of genetic diversity. Specifically, parasites displaying complex life cycles, such as digeneans, were more abundant for the medium IR level. This was also reflected in a higher mean value for the LCI.

The observed population pattern (i.e., percentages of individuals) in the present study could be in accordance with a scenario of disruptive selection, i.e., leading to a bimodal trait distribution [23,81]. In particular, a negative selection appeared to mainly focus on individuals displaying intermediate genetic diversity, with the medium level being less common, whereas the most homozygous and heterozygous pipefish would be positively selected (i.e., extremes being more frequent). Mechanistically, the selective pressure promoting such disruption within this pipefish population may be related to a differential effect of parasitism, depending on host genetic traits, as suggested per Blanchet et al. [26,27]. However, such differences among population frequencies were not (statistically) clear in the present study and, consequently, these should be considered a trend but not a real effect. A potential explanation for this lack of statistical significance may be that examined specimens belonged to a particular fraction of the entire population: the oldest adults, as the largest specimens, were analysed, and fish typically exhibit continuous growth throughout their full life-span. In the Iberian Peninsula, clear ecological patterns relating a variety of parasitological traits were observed after using larger/older specimens to assess colonisation capacity of a minnow population [51,57]) and as ecological indicators by using pipefish and goby species [39]. Moreover, Blanchet et al. [26] found no effect (i.e., statistical interaction) of host body size on any mathematical model including IR and parasite load for a French dace population. These previous studies provided the rationale to select the largest individuals of pipefish in the Mar Menor. However, juvenile fish may be more vulnerable to parasitism and, consequently, better reflect this effect on populations under certain environmental circumstances, such as heterogeneity of habitat features [53,82]. Thus, future research should focus on a broader size range to overcome this potential limitation. Overall, this more comprehensive analysis would truly reveal the differential effect of parasites depending on genetic diversity at the entire population level: juvenile, sub-adult, and adult individuals. Additionally, the assessment of genetic markers in juvenile pipefish could also shed light on the effect of parasites on population recruitment, a key trait to assess viability of fish communities on a long-term basis [4,83].

Parasite prevalence and abundance usually follow the same variation trends, as observed by Chapman et al. [53] in a freshwater fish (pumpkinseed sunfish *Lepomis gibbosus*). Similarly, these parasitological parameters showed the same pattern across the genetic categorisation, with both more homozygous and heterozygous individuals harbouring, on average, fewer parasites than pipefish from the intermediate level of genetic diversity. Despite significant differences not being clearly detected for total prevalence and abundance, such results do merit some consideration in concordance with endo-parasite data, as again they could be understood through a mechanism of disruptive selection. Thus, this was contrary to the expectation on directional selection: a lower parasite burden only for the most heterozygous individuals [24,84], which has also been observed in different fishes, such as Atlantic salmon [25] and guppy *Poecilia reticulata* [85]. In this genetic context where heterozygosity is highly advantageous, balancing selection (i.e., overdominance of heterozygous specimens) could also occur at a certain degree of magnitude (e.g., see Mäkinen et al. [86] for a study on three-spined stickleback *Gasterosteus aculeatus*), which would contribute to the maintenance of the same level of genetic diversity over time [87]. Alternatively, Blanchet et al. [26] found that disruptive selection was acting on a dace population infected by a fin-feeder parasite by using both continuous data and categories of genetic diversity [27]. As per these authors, a similar genetic/parasitological pattern was observed by using IR categories, prevalence percentages and abundance values from pipefish from the Mar Menor, i.e., explained as disruptive selection. These overall findings are very relevant, as this same evolutionary trend has been observed on very contrasting species, in terms of phylogenetic distance and ecological requirements, such as a leuciscid fish in fresh waters and a syngnathid species in transitional waters. Indeed, parasitological examinations were much less complete in the mentioned studies: one helminth species in Zueva et al. [25], two species from the same Genus in Fraser and Neff [85], and one copepod species in Blanchet et al. [26,27]. However, a higher variety of parasites were analysed in the present work: protozoans, cestodes, monogeneans, digeneans, nematodes, acanthocephalans and crustaceans. Such an assessment of several parasite taxa at the same time is obviously more representative, but it poses a risk for biological interpretation, as they can display a wide range of traits (e.g., virulence, transmission) that may differentially affect interactions with host individuals. Therefore, this elevated variability could be the reason why statistical differences did not reach the significance level for total prevalence and parasite abundance in pipefish, potentially masking the underlying ecological and evolutionary processes. This wider taxonomic variety could also be related to the lack of statistical significance for any continuous relationship between both tA and endoA with genetic diversity (see Section 2.6). In the case of Blanchet et al. [26], these authors did find a significant quadratic regression between abundance and IR, probably because they only focused on a single parasite species (*T. polycolpus*), which displayed a deep impact on host health (damage of fins).

With respect to health status, BC and HAI have previously been used to detect variations in physical fitness of fishes inhabiting transitional waters [39]. Both morphological parameters exhibited the same pattern, with pipefish individuals that are either more homozygous or more heterozygous displaying a better condition/health. These findings were in accordance with the parasitological pattern, likely related to the effect of host genotypes on susceptibility to infection and parasite transmission (see a detailed explanation below). However, a statistically significant difference across genetic categories was only found for HAI. In the specific case of BC, this index did not reach the significance level and, consequently, it should be considered a trend but not a real effect. Indeed, other factors not related to parasites may also modulate fish condition. In particular, indices calculated from data on body mass are very sensitive to variations in food supply [55] and this may affect results on BC. The pipefish species selected for this study chiefly feeds on copepods and cladocerans, two important biological communities within zooplankton in Iberian waters [50,88]. These crustacean food items are not usually a limiting factor in the Mar Menor because environmental conditions can periodically change from an oligotrophic status to a eutrophication level. Such episodes are unfortunately common within this coastal lagoon, which is of conservation concern [89]. Consequently, this bottom–up effect promotes an increase in phytoplankton, with the corresponding biomass of zooplankton grazers [90]. Statistically speaking, HAI was a good indicator of fitness status for pipefish in relation to genetic diversity, with higher values being found in the medium category: more parasites and worse health. In this respect, Chapman et al. [53] observed that this same index could reflect physiological/histological effects of infectious diseases via parasitism on gills and guts by ecto-/endo-helminths, with positive correlations being found between HAI scores and parasite loads. Although a significant correlation does not always imply causation, parasitism may directly impact fish organs while feeding, which promotes a variety of histo-pathological effects, such as tissue inflammation, hyperplasia, hypertrophy and necrosis [91]. The closer connection with these tissue damages was likely the reason why this health index was able to provide clearer differences between levels of genetic diversity. Such findings demonstrate the importance of using both external and internal indices of fish health status, with different levels of sensitivity, to better detect variations across genetic, morphological and parasitological traits [57].

According to Råberg et al. [21], hosts have evolved two different strategies to maintain fitness when interacting with parasites and pathogens: (1) resistance, which would be the ability to actively limit and reduce parasite burden before or after infection has occurred, and (2) tolerance, which would be the ability to actively limit and reduce the damage caused by a given parasite burden. These authors also demonstrated how both strategies deeply depended on genetic variation, i.e., displaying different reaction norms across host genotypes. It follows, then, that tA and mainly endoA can be considered proxies for resistance, whereas HAI can be considered a proxy for tolerance in the present study. Overall, both mechanisms of resistance and tolerance may act in a synergistic manner, decreasing the prevalence and abundance of diseases, as well as maintaining health, for the highest levels of homozygosity and heterozygosity within this pipefish population. Contrary to this interpretation, Råberg et al. [21] and Blanchet et al. [27] provided evidence that a negative genetic correlation existed between resistance and tolerance, such that a highly tolerant individual is necessarily weakly resistant (and vice versa). Again, it must be clearly stated that correlation does not imply causation. Focusing particularly on the ‘fish’ study, which is the most equivalent to the present work [27], the level of fin degradation was the only trait used in common dace as an inverse descriptor of host health for a given number of parasites (i.e., tolerance). Nevertheless, the assessment of health status was more comprehensive in pipefish from the Mar Menor, with each individual being examined for up to eight anatomical regions, including internal organs (see the Section 2.3). This could be a potential explanation for finding the same clear trends for endoA and HAI values, which are inverse for resistance and tolerance, respectively, across IR intervals in this study.

Regarding the genetic basis of these two physiological processes, specific genes and genomic regions have been related to the immunological capability for resisting parasites at the molecular (e.g., immuno-globulins) and cellular (e.g., lymphocytes, macrophages) levels [25,92,93]. In relation to tolerance, particular host genotypes are better able to control an exacerbated immune (inflammatory) response by acting on cytokines or mast cells [92,94], as well as reducing tissue degradation through investment in cell regeneration [22]. Following this genetic perspective and according to the immunological hypothesis, research on polymorphic *loci* coding for the Major Histocompatibility Complex (MHC, hereafter) is promising, given their important role in the adaptive immune response of vertebrates: presenting pathogen-derived antigenic peptides to lymphocytes [3]. Thus, an elevated diversity of these genes (via heterozygosity) is considered a good indicator of resistance in host fishes [84,95]. Within the study pipefish population, the present results appear to reveal two well-defined and efficient contrasting strategies against parasites at the host genotypic level: (1) more heterozygosity, and (2) more homozygosity. The first one is easy to understand, according to previous observations [24,95], where an elevated genetic diversity is surely reflected in a wider repertoire of immune response. This promotes a rapid detection on a great variety of parasites and pathogens (antigens), and hence hosts can readily eliminate them [25,95]. The second strategy is less intuitive, but a potential explanation could be based on the polygenic nature of the immune response [25,30,96]. A large number of allelic copies encoding for particular products (e.g., antibodies, cytokines, MHC molecules), although less diverse due to homozygosity, could reach a higher synthesis rate (i.e., a gene–dose effect; [97]), with this also being an effective mechanism to control parasites/pathogens. As an example, Spiering and de Vriesh [98] observed a higher bi-allelic expression/synthesis against COVID-19. For the medium category, these pipefish individuals would be neither diverse nor productive enough to display a suitable level of immuno-competence, facilitating penetration/migration of parasites through host tissues and, consequently, being more susceptible to infections. Such a (worse) genotypic context was also reflected in the highest HAI mean (i.e., poor health status).

Parasite taxa recovered from pipefish were similar to those found in previous research on the same fish species, although richness was lower [64,99]. This finding (i.e., a lower richness) may be due to the particular hypersaline conditions within the Mar Menor lagoon (see description of the Study Area above). In terms of poor water quality, elevated salinity commonly affects fish parasites via direct (e.g., ciliated and free-swimming larvae, such as miracidium searching for mollusc hosts) and indirect effects (e.g., host tolerance) [100]. In the present study, pipefish belonging to the low/high levels of genetic diversity were able to avoid infections from three parasite species (two digeneans and one acanthocephalan), as well as maintain lower parasite abundances and LCI values, with respect to the medium category. For the low/high categories, monoxenous ecto-parasites (skin and gills) were more abundant, such as *Ergasilus ponticus* (crustacean), *Gyrodactylus* (monogenean) and *Trichodina partidisci* (ciliate), which display direct cycles, i.e., they are transferred from fish to fish. The acanthocephalan *Pomphorhynchus laevis* was also more frequent in pipefish intestines from those two categories. This helminthiasis needs two species to complete its heteroxenous life cycle: amphipods (intermediate crustacean host) and fish. Therefore, all these parasite taxa established a common link with pipefish, which always played a role as the unique or definitive host: where sexual reproduction takes place by the adult stage. Thus, fish usually remained alive after this parasitic interaction [8]. Yet, parasitism was very different for pipefish from the medium category, where all the observed digeneans (including the most abundant parasites: Cyathocotylidae, *Podocotyle*, *Cryptocotyle* and *Diplostomum*), as well as the nematode *Contracaecum microcephalum* and the cestode *Proteocephalus*, were more frequent. All these taxa display heteroxenous cycles. Except for *Podocotyle atherinae* and *Proteocephalus* sp. (with small zooplanktivorous fish being the definitive host), the rest of those helminths ‘used’ pipefish as an intermediate host that must be preyed upon by larger predatory fish (e.g., sea bass *Dicentrarchus labrax*) or mostly aquatic birds (e.g., cormorants, gulls, herons and egrets) as the definitive hosts [101]. These piscivorous species are common in the study area (see Oliva-Paterna et al. [41] for fishes and Farinós-Celdrán et al. [102] for birds in the Mar Menor) and they usually prey on pipefish species [103,104]. In fish hosts, encysted metacercaria larvae can shelter in the muscle/skin (*Cryptocotyle concavum*), eye lens (*Diplostomum* sp., causing impaired vision) and other vital organs (Cyathocotylidae). This parasite stage is aimed at changing fish behaviour to become more vulnerable to predation by piscivorous definitive hosts. As an example of this, such parasitised fish usually spend more time swimming near the surface [105]. In the case of *Contracaecum microcephalum*, definitive hosts are marine mammals, such as seals and cetaceans. For this nematode species, fish play a role as an intermediate or even more specifically, as a paratenic host, which is a non-essential species for the parasite cycle, although it can increase the transmission rate (similarly to the phylogenetically related and well-known Genus *Anisakis*; [106]). These kinds of parasite species (digeneans and nematodes) were more common in the medium category, where such infestations of those anatomical systems/organs may cause severe tissue damage. Thus, a stronger pathological effect would ultimately be necessary for these parasites to provoke fish death via predation by definitive hosts. Such negative pressure on pipefish from the medium level of genetic diversity was likely reflected in higher HAI values (see above), promoting clear statistical differences. These findings on parasite composition were directly related to the LCI and endoA results, with averages significantly higher for the medium category. This indicated an elevated abundance of heteroxenous parasites that need several ecologically diverse hosts (e.g., zooplankton, snails, fish, birds or mammals) to complete their life cycles. Thus, endo-parasites inhabiting internal organs can avoid the harsh environmental conditions outside the fish body, such as skin and gills for ecto-parasites, promoting a deeper impact on host tissue health. Apparently, pipefish from the low/high categories were capable of better resisting (i.e., stronger immune response) those more noxious parasites, i.e., heteroxenous cycles, which necessarily implies fish mortality [101].

## 5. Conclusions

This study is underpinned by the use of particular traits to evaluate the intimate ecological and evolutionary relationships between parasitism and host genotypes in animal populations. Thus, all the analysed parameters in the present work (i.e., frequencies of host population, health indices and parasitological traits) always provided patterns that could be easily explained under a scenario of disruptive selection. Overall, the results suggests that this type of natural selection has been occurring in pipefish inhabiting the Mar Menor coastal lagoon. However, this conclusion should be tentative, as such associations do not necessarily imply causation. Indeed, it must be highlighted that the clearest differences (statistically speaking) were revealed for HAI (a more internal index), along with LCI and endoA (both indicators of more noxious parasites). These results were probably because such parameters reflect deeper impacts on hosts (see comments above). From a strict micro-evolutionary perspective, parasites appear to play a crucial role for shaping genetic diversity within local communities of animal hosts. In particular for the study pipefish population, genetic variability (or variance, in mathematical terms) was broadened through disruptive selection, which had already been observed by Blanchet et al. [26,27] in a very different fish species: the leuciscid common dace inhabiting fresh waters. Therefore, this study demonstrates that parasite-mediated disruptive selection is a widespread phenomenon across finfish populations, as well as other wild aquatic communities (see the work of Duffy et al. [107] for an example on a crustacean species from zooplankton). Also, the present results complement the most intuitive view that a high heterozygosity is the only possible means to achieve a fitter status [24,25,84], with two main strategies being displayed at the host genotypic level to better deal with parasites: (1) high homozygosity (gene–dose effect on immune synthesis), and (2) high heterozygosity (wider repertoire of immune response). Mechanistically, homozygosity level (i.e., genotypic variability measured by the IR) may reflect immuno-competence as a phenotypic trait [108], with parasites exerting a higher negative (purifying) selection on host individuals from the intermediate range, displaying a poorer immune response. These findings open new perspectives for future research lines on the genetic interaction between hosts and their parasites, as well as the underlying processes. Consequently, researchers from different academic disciplines, such as animal health, ecology and evolution, may find the present study a valuable contribution to understand the genetic mechanisms by which hosts adapt to pathogen/parasite infections.

## Figures and Tables

**Figure 1 animals-15-02195-f001:**
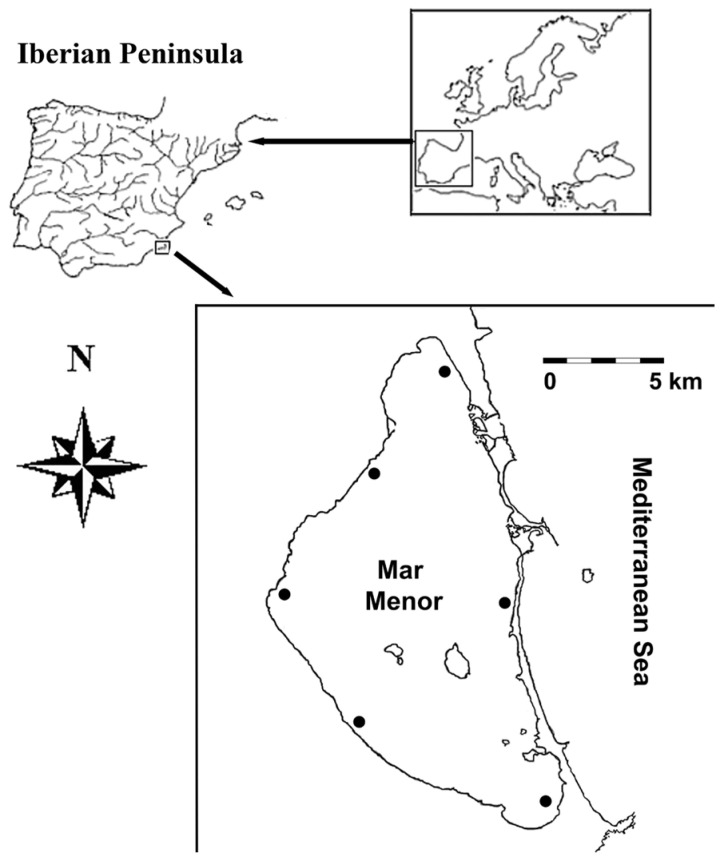
Map of the study area showing the geographic location (coordinates: 37°45′ N–0°47′ W) for the Mar Menor coastal lagoon (SE Spain). Black dots: sampling sites.

**Figure 2 animals-15-02195-f002:**
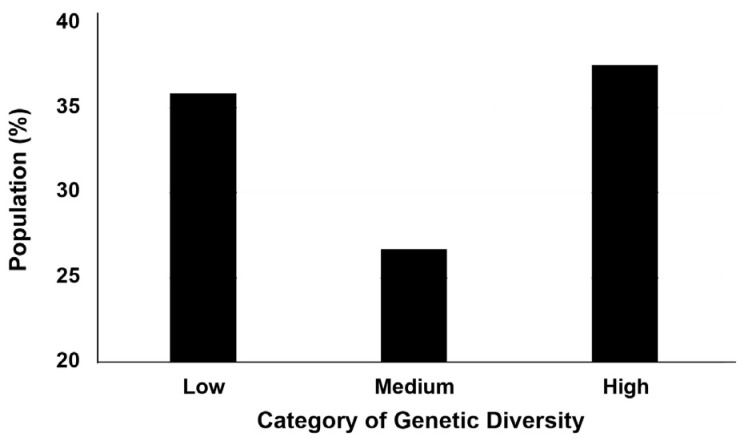
Population frequencies for black-striped pipefish *Syngnathus abaster* from the Mar Menor coastal lagoon. The results are percentages of fish individuals, reported per category (level) of genetic diversity.

**Figure 3 animals-15-02195-f003:**
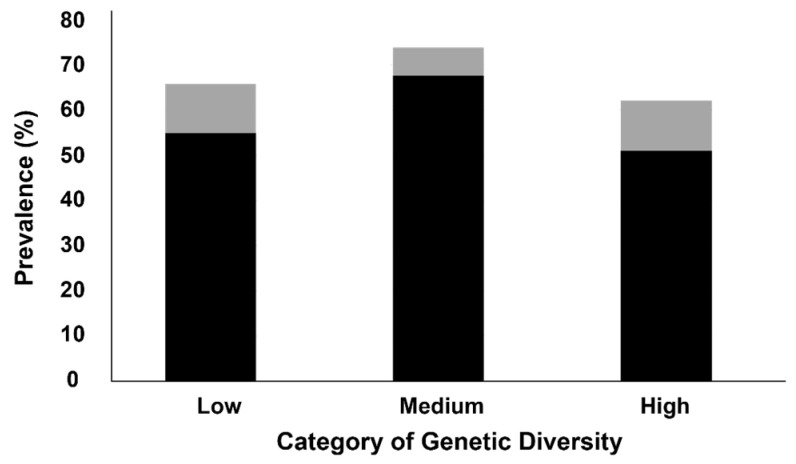
Total parasite and endo-parasite prevalences for black-striped pipefish *Syngnathus abaster* from the Mar Menor coastal lagoon. The results are percentages of parasitised fish individuals, reported per category (level) of genetic diversity. Black colour: endo-parasites detected; grey colour: only ecto-parasites detected.

**Table 1 animals-15-02195-t001:** Description of the microsatellites (*n* = 10) used in this study to estimate genetic diversity in black-striped pipefish *Syngnathus abaster* from the Mar Menor coastal lagoon. For each pair of primers (F-Forward, R-Reverse), the following information is provided: *locus* name and GenBank ID, specific sequences, fluorophore name (F/R-attached), allelic size range (Size), allelic richness (N_A_), core motif and bibliographic reference.

*locus* (GenBank ID)	Primers 5′–3′	Size	N_A_	Core Motif	References
Sabas1 (GQ168557)	F-FAM: GGCATGTATCAAACGAATGTCCTC	147–451	28	(ATCT)_32_	[67]
	R: TTTGCGAAACAGTGTATGACTATACTT				
Sabas2 (GQ168558)	F-ATTO565: CAGGTTGTTGACCATTTGAGTGT	170–262	11	(TC)_33_	[67]
	R: CGAATATGATTATGTGAGTCCTAAGGC				
Sabas3 (GQ168559)	F: TTCCCCCTAGGACCAATAAAGTATCT	149–279	21	(ATCT)_37_	[67]
	R-Bodipy530: TGAGAGTGGTTGCCTCCAGC				
Sabas4 (GQ168560)	F: CAAAATGCAAGTGATCCTGTGTAGG	207–298	27	(TCTA) _38_	[67]
	R-ATTO550: TGGTGTGGTGGAACTGAATGACG				
Sabas5 (GQ168561)	F: CATTGAAACTGCATTGATTTTATGATT	249–360	29	(ATCT)_44_	[67]
	R-FAM: AGGGGGTTGTAAGTCTTTGTG				
Sabas6 (GQ168562)	F-ATTO565: TCGTGTTCCGGGACGCACATGG	244–451	35	(AGAT)_4_CTAT(ATCT)_31_	[67]
	R: ATGTCCGAGGTCAAACACGGCGA				
Sabas7 (GQ168563)	F-Bodipy530: CGATGTGCGAGACCTGTTGCG	184–322	24	(GATA)_33_	[67]
	R: AAAGAGGCGGAGCTTGTGTAAGGA				
Sabas8 (GQ168564)	F: FAM-TATGTGTGCCCTGCGACTGGTTG R: CAGGAGATAAGGGAGCGTTTATAGCGG	187–522	31	(TAGA)_42_	[67]
Sabas9 (GQ168565)	F: TGATTTGGAATGACACGGGTGGTTTG	200–276	17	(ATAG)_33_	[67]
	R-FAM: TCGTTTTGTGTGCACCGAGTGTT				
typh16 (–)	F-ATTO565: CAGGACACGCTGGAAAGAC	223–307	20	(GATG)_15_	[68]
	R: GCAACACCTTGAAGAGGAAAGT				

**Table 2 animals-15-02195-t002:** Morphological traits of health status (BC and HAI) for black-striped pipefish *Syngnathus abaster* from the Mar Menor coastal lagoon. The results are (adjusted) means ± SD, reported per category (level) of genetic diversity. *F*-ratios, degrees of freedom and significance levels (*p*-values) are presented, after ANCOVA (covariates: SL/eBM, see Data Analyses for details). Lower-case superscript letters indicate significant differences between categories (Tukey’s HSD test: *p* < 0.05).

S. *abaster*	Low	Medium	High	*F* _2,356_	*p*-Value
BC (mg)	411.5 ± 200.90	314.1 ± 190.85	402.5 ± 211.62	2.36	0.095
HAI (points)	^b^ 24.2 ± 13.16	^a^ 35.3 ± 14.07	^b^ 25.4 ± 15.55	4.95	0.008

**Table 3 animals-15-02195-t003:** Parasite list found in black-striped pipefish *Syngnathus abaster* from the Mar Menor coastal lagoon. Mean values of parasite abundances are shown per Family, Genus or Species, and category (level) of genetic diversity. Dashes indicate parasites were undetected on examined pipefish individuals (*n* = 85, 71 and 84 for the low, medium and high levels of genetic diversity, respectively).

Parasite (Taxon)	Parasite (Family/Genus/Species)	Low	Medium	High
Ciliophora	*Trichodina partidisci*	0.7	0.3	0.7
Cestoda	*Proteocephalus* sp.	0.3	0.5	0.4
Monogenea	*Gyrodactylus* sp.	2.2	0.6	2.9
Digenea	*Cryptocotyle concava*	1.5	2.8	1.2
	Cyathocotylidae	3.8	7.3	4.4
	*Diplostomum* sp.	1.3	2.2	1.5
	*Echinochasmus perfoliatus*	–	1	0.2
	*Podocotyle atherinae*	2.7	3.9	2.5
	*Timoniella imbutiformis*	0.9	1.7	–
Nematoda	*Contracaecum microcephalum*	0.1	0.8	0.3
Acanthocephala	*Paracanthocephaloides incrassatus*	–	0.3	0.6
	*Pomphorhynchus laevis*	1.3	0.2	1.1
Crustacea	*Ergasilus ponticus*	2.9	1.3	2.7

**Table 4 animals-15-02195-t004:** Parasitological traits (tA, LCI and endoA) for black-striped pipefish *Syngnathus abaster* from the Mar Menor coastal lagoon. The results are (adjusted) means ± SD, reported per category (level) of genetic diversity. *F*-ratios, degrees of freedom and significance levels (*p*-values) are presented, after ANCOVA (covariate: eBM, see Data Analyses for details). Lower-case superscript letters indicate significant differences between categories (Tukey’s HSD test: *p* < 0.05).

*S. abaster*	Low	Medium	High	*F* _2,236_	*p*-Value
tA (number of specimens)	17.7 ± 8.68	22.9 ± 9.37	18.5 ± 8.36	2.59	0.077
LCI (range: 1–3 host species)	^b^ 1.57 ± 0.685	^a^ 1.99 ± 0.701	^b^ 1.56 ± 0.806	3.92	0.021
*S. abaster*	Low	Medium	High	*F* _2,201_	*p*-Value
endoA (number of specimens)	^b^ 12.2 ± 8.28	^a^ 20.7 ± 9.01	^b^ 11.9 ± 8.13	3.81	0.024

## Data Availability

Data are available from the authors upon reasonable request.

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
