# Peer review of "Effects of Genetic Diversity on Health Status and Parasitological Traits in a Wild Fish Population Inhabiting a Coastal Lagoon"

_animals, 2025, doi:10.3390/ani15152195_

Round 1
Reviewer 1 Report
Comments and Suggestions for Authors
The present work intends to establish a relationship among fish health, parasite burden and genetic diversity based on specimens of Sygnathus collected in a Spanish coastal lagoon. Althoug this topic is of undeniable scientific interest, I believe that the work adduces major flaws in conceptualization and execution and I suggest rejection for publication.
Although I do not deny that a disruptive selection may be in action in the case of the target species, any evidence is provided as regards this may be related to fish parasitology or health status in the present case.
Authors recognize in the discussion that their results do not allow concluding a differential effect of parasitism among fish groups depending on their genetic diversity. Indeed, a relationship between parasites and genetic diversity is not established. This would not be a big problem and is not the reason of my suggestion of rejection of this work, since a negative outcome can also be an interesting result woth of being discussed and published. I truly believe that the majer concern in the present case is a problematic conceptualization, and a poor experimental / field design.
Conceptually, parasite assemblages and genetic diversity operate at different temporal scales. While the genetic composition of an individual remains constant through its life, its parasite community varies as it grows due to ontogenic shifts, seasonally as a result of a range of factors like environmental variability, different availability of intermediate hosts, and changes on the local free-living communities. On the geographical scale, both parasites and genetic composition can inform on host populations differentiation and can be correlated in some cases and if correctly assessed, but parasites also display high variability at the local scale (as reported for the target species in previous and in the present study). Present data do not come from different fish populations, and the study is intended at the intrapopulation level, for which natural variability can obscure relationships among variables operating in the longer term and in broader scales. Not surprisingly, no relationships among parasites and genetic diversity are reported.
General fish condition indices, which have a complex respondse to multiple causative and modullating agents, do not allow to establish a direct link between health and genetic diversity. I wish it was so simple! I'va also used for years these general health indices in fish, and they can rarely be associated to a specific factor unless a very determining agent with a strong effect is involved in the system studied (e.g. a very virulent parasite). Furthermore, some of these indices are not exactly indicators of fish health: general body condition indices mostly vary in response to the fish resproductive cycle, for instance.
I will skip detailed comments here becuase I think that the major flaw of this work is a very general conceptual issue. However, the conceptual link between genetics and fish health and parasites is not clear. Which of these is considered by the authors to exert effects on the other? Contradictory statements can be found throughout the manuscript in this sense. The discussion is rather a review of the available bibliography than a discussion of own results, because these clearly do not allow for an elaborate discussion.
In conclusion, a significant improvement in the general design of the study should be carried out. Provided this is done, I agree with the authors in that the study could be improved by broadening the scope of the sampling. Correlation in many cases does not imply causation, and if the hypothesis is that it does, a thorough and accurate sampling and experimental design must be established to prove it. This is, unfortunately, not the case. This is why I suggest rejection of the present manuscript.
Comments on the Quality of English LanguageThe use of English is generally correct and is thus not a major problem in this case, although the ms would benefit for a revision at this respect. An important aspect I would like to highlight: I urge authors that for future works avoid the constant use of quotation marks (mostly unnecessary in the present case) and of parenthesis, which difficult the reading of the text.
Author Response
Dear Reviewer 1, I attached a document with responses.
Regards,
David Almeida and co-authors

Reviewer 2 Report
Comments and Suggestions for Authors
The research was made on sufficient material and with the use of adequate laboratory procedures and statistical analysis. The authors carefully planned and described the collection of material, which is very important for statistical comparisons, since different conditions (period and location of material collection, sex, age, reproductive status) could influence the direction of selection.
The authors have disclosed and discussed all potential limitations of this study in the Discussion section. Perhaps the use of all parasite species (13) in the model did not allow the authors to identify clearer statistical differences in a number of cases (masked them), but it more fully reflects the real complex situation in the wild.
The manuscript was prepared with high quality and did not raise any questions or misunderstandings when reading it.
The originality of the manuscript is 66%, self-citations is 31%. The bibliography is extensive and corresponds to review articles. But I am inclined to regard this as an advantage of the paper. Moreover, I didn't find any inappropriate citations.
Although I am not a native speaker, the English was clear and easy to read.
The text of the manuscript was carefully prepared, I didn't find any errors. Minor typos were noticed in the bibliography, for example, in source number 31 (Line 1011).
Author Response
Dear Reviewer 2, I attached a document with responses.
Regards,
David Almeida and co-authors

Reviewer 3 Report
Comments and Suggestions for Authors
The study used external (Body Condition, BC) and internal (HAI) health indices. ​ The results are very meaningful and valuable, but in my opinion it is necessary to separate ectoparasites from endoparasites for further calculations. If you consider that fish live in areas that are influenced by both salt and fresh water, this is particularly reflected in the ectoparasites. Endoparasites live in stable conditions of the internal environment and cannot be treated in the same way as ectoparasites. I therefore suggest separate tables and separate graphical solutions as well as a reformulation of the discussion
Author Response
Dear Reviewer 3, I attached a document with responses.
Regards,
David Almeida and co-authors

Reviewer 4 Report
Comments and Suggestions for Authors
Overall, the paper was written well, and the English was good. Thank you for the work you put forth into the writing.
The methods and results are difficult to follow. The authors were trying to be concise and brief with their writing, which is appreciated, but not enough details were provided within the actual manuscript to determine what exactly happened with methods. The citations were good to back up the work and to show that others have done work before but there needs to be more details of what the authors did. As the methods are currently written it would be difficult to reproduce the majority of the research. The genetics method section was well written.
Please see my comments for the results section. It is unclear why the authors present some of the statistical testing based on the p-values.
Because of the difficulty to determining what was going on with the methods and the results I did not fully read the discussion. It would be helpful for a first paragraph to simply recap the important results and how genetic diversity and “Health Status” are linked etc.
Please see attached document for detailed comments.

Author Response
Dear Reviewer 4, I attached a document with responses.
Regards,
David Almeida and co-authors

Round 2
Reviewer 4 Report
Comments and Suggestions for Authors
I would like to thank the authors for the time they took to address the comments through the review process. Below are my minor comments. Well done.
Line 413: Please provide brief reason to why log-transformation was used.
Line 413: Please revise to “A Shaprio-Wilks test…”
Lines 455 - 457: What is complexity of GAM? I am unfamiliar with the wording of “a stepwise procedure according to Akaike information criterion.” Please provide a citation for the use of this methodology. Reading the sentence in Line 457-458 it seems you are just using normal AIC methodology. This could be written as: “The highest supported GAMs were selected using Akaike information criterion (Burnham and Anderson 2002)”.
Lines 470-511: Thank you for addressing my comments about specific information and the models used.
Lines 646-661: Thank you for addressing my comment about a recap paragraph.
Author Response
Dear Reviewer,
I document with responses has been attached.
Regards,
David Almeida and co-authors.
